# The Impact of Chemotherapy and Transforming Growth Factor-β1 in Liver Regeneration after Hepatectomy among Colorectal Cancer Patients

**DOI:** 10.3390/jpm14020144

**Published:** 2024-01-28

**Authors:** Rokas Račkauskas, Raminta Lukšaitė-Lukštė, Rokas Stulpinas, Augustinas Baušys, Marius Paškonis, Mindaugas Kvietkauskas, Vitalijus Sokolovas, Arvydas Laurinavičius, Kęstutis Strupas

**Affiliations:** 1Clinic of Gastroenterology, Nephrourology, and Surgery, Institute of Clinical Medicine, Faculty of Medicine, Vilnius University, Ciurlionio Str. 21, LT-03101 Vilnius, Lithuania; 2Department of Radiology, Nuclear Medicine and Medical Physics, Institute of Biomedical Sciences, Faculty of Medicine, Vilnius University, LT-03101 Vilnius, Lithuania; 3Department of Pathology, Forensic Medicine and Pharmacology, Vilnius University, LT-03101 Vilnius, Lithuania; 4National Center of Pathology, Affiliate of Vilnius University Hospital Santaros Clinics, LT-08406 Vilnius, Lithuania

**Keywords:** liver resection, regeneration, cancer, metastases

## Abstract

An ongoing debate surrounds the impact of chemotherapy on post-hepatectomy liver regeneration in patients with colorectal cancer liver metastases (CRLM), with unclear regulatory mechanisms. This study sought to delve into liver regeneration post-resection in CRLM patients, specifically examining the roles of hepatocyte growth factor (HGF) and transforming growth factor β1 (TGF-β1). In this longitudinal observational study, 17 patients undergoing major liver resection for CRLM and 17 with benign indications as controls were enrolled. Liver regeneration within 30 postoperative days was assessed via CT, considering clinicopathological characteristics, liver enzymes, liver stiffness by elastography, and the impact of HGF and TGF-β1 on liver regeneration. The results revealed that the control group exhibited significantly higher mean liver regeneration volume (200 ± 180 mL) within 30 days postoperatively compared to the CRLM group (72 ± 154 mL); *p* = 0.03. Baseline alkaline phosphatase (AP) and TGF-β1 blood levels were notably higher in the CRLM group. Immunohistochemical analysis indicated a higher proportion of CRLM patients with high TGF-β1 expression in liver tissues compared to the control group (*p* = 0.034). Correlation analysis showed that resected liver volume, baseline plasma HGF, AP, and albumin levels significantly correlated with liver regeneration volume. However, in multivariable analysis, only resected liver volume (β: 0.31; 95% CI: 0.14–0.47, *p* = 0.01) remained significant. In conclusion, this study highlights compromised liver regeneration in CRLM patients post-chemotherapy. Additionally, these patients exhibited lower serum TGF-β1 levels and reduced TGF-β1 expression in liver tissue, suggesting TGF-β1 involvement in mechanisms hindering liver regeneration capacity following major resection after chemotherapy.

## 1. Background

The colorectal cancer (CRC) incidence rate is growing. It is the third most common and second deadliest malignancy worldwide, with more than 1.9 million new cases and 935,000 cancer-related deaths annually [1]. Recent substantial progress in both diagnostic and therapeutic techniques has enabled the early detection and treatment of this disease [2]. In spite of these enhancements, every second CRC patient develops metastases, and the liver is the most common site [3]. The treatment of colorectal cancer liver metastases (CRLM) requires a multidisciplinary approach in order to apply complex strategies that encompass the synergistic application of systemic and local therapies. However, only local treatment, such as surgical resection, may be curative. Despite the progress made in surgical techniques, the critical constraint in liver resection persists in the form of insufficient remnant liver volume and the subsequent risk of potentially lethal postoperative liver failure [4]. The current recommendation advocates preserving a minimum of 30% of the liver volume post-resection to avoid post-hepatectomy liver failure in patients treated with chemotherapy [5,6]. The liver is unique among solid organs in utilizing regenerative mechanisms to maintain its liver-to-bodyweight ratio consistently at 100%, aligning with the requirements for body homeostasis [7]. A substantial number of CRLM patients, previously categorized as unresectable, can now undergo complete resection due to the implementation of innovative strategies that facilitate the manipulation of liver volume (4). However, chemotherapy’s impact on liver regeneration remains controversial, with conflicting evidence existing [2,8,9]. A healthy liver rapidly restores its mass, particularly within the initial week following a major resection. Subsequently, the pace of regeneration gradually decreases, ultimately ceasing between 6 months and 1-year post-hepatectomy (7). It is crucial to assess the impact of preoperative chemotherapy on liver regeneration, especially when planning a second or repeated hepatectomy. Biomarkers that would predict postoperative liver regeneration would be an option to personalize surgery and further improve CRLM treatment outcomes, although none are available currently. Several studies have investigated hepatocyte growth factor (HGF) and tumor growth factor β1 (TGF-β1), both recognized for their influence on liver volume regeneration and considered as pro- and anti-regenerative indicators, respectively [10,11]. Although, despite the fact that these potential biomarkers for post-hepatectomy liver regeneration are broadly investigated in preclinical research, there is a lack of evidence from clinical studies. Therefore, the objective of this study was to explore liver regeneration in patients undergoing liver resection for CRLM after chemotherapy, focusing on the roles of HGF and TGF-β1 in this process.

## 2. Materials and Methods

### 2.1. Ethics

This longitudinal observational study was conducted between 2019 and 2023 at Vilnius University Hospital Santaros Klinikos, Vilnius, Lithuania. The study commenced after the protocol had received approval from the Vilnius University Regional Bioethics Committee (No. 2019/3-1112-605) and was conducted in accordance with the ethical standards of the Helsinki Declaration of 1975. All the patients provided written informed consent before participating in the study.

### 2.2. Participants and Data Collection

Patients aged between 18 and 90 years who were scheduled to undergo ≥2 segments liver resection for CRLM after treatment with systemic chemotherapy or for benign disease were eligible for the study. Clinicopathologic and treatment data were collected; static liver function tests, elastography, and abdominal computed tomography (CT) were performed to evaluate liver stiffness and post-hepatectomy regeneration; and an enzyme-linked immunosorbent assay (ELISA) and immunohistochemistry (IHC) were performed for HGF and TGF-β1. For a detailed description of the experiments, see below. Synchronous liver metastases were defined as those detected at the time of presentation of the primary CRC, while liver metastases detected after diagnosis of the primary tumor, but absent at presentation, were defined as metachronous metastases [12].

### 2.3. Calculation of Liver Volume by CT

Abdominal CT was performed before liver resection (baseline) and on postoperative days (POD) 1 and 30. CT scans were performed using the GE Discovery 750hd (128 slices) system (GE HealthCare, Chicago, Illinois USA). Liver volume was calculated using Philips IntelliSpace Portal Image and Information Management Software, Liver volume, and segmentation analysis package, Version 12.1 2020 (Philips Medical Systems, The Netherlands).

### 2.4. Liver Stiffness Analysis

Liver ultrasound elastography 2D-SWE (Canon Aplio i800 system, Canon Medical Systems Corporation, Otawara, Tochigi, Japan) was performed on the baseline, POD 1, and POD 30 using a convex probe in a supine position with the right arm elevated above the head. Measurements were performed in the right liver lobe, avoiding major blood vessels, and at least 1 cm away from the liver capsule. Ten measurements were performed in each patient, and the results were expressed in kPa. Measurements within the interquartile range to the median range (IQR/M) of less than 30% were accepted as being valid. The 2D-SWE imaging protocol was set according to Barr et al. [13].

### 2.5. ELISA and Immunohistochemistry for HGF and TGF β1

Serum samples were collected on the day before surgery and POD 1, 7, and 30. ELISA assays were carried out using HGF (HGF Human Instant ELISA Kit, 128 Tests, Thermo Fisher Scientific, Waltham, Massachusetts, USA) and TGF β 1 (TGF β 1 Human Instant ELISA Kit, 128 Tests, Thermo Fisher Scientific, Waltham, Massachusetts, USA) antibodies according to the manufacturers’ instructions.

For immunohistochemistry, 3 µm paraffin-embedded sections of non-tumorous liver tissue from resected specimens were used. IHC for TGF β 1 and HGF was performed with a multimer-technology-based detection system, ultraView Universal DAB (Ventana, Tucson, AZ, USA). The TGF β 1 antibody (Rb mAb TGF beta 1 [EPR21143]; Abcam, Cambridge, UK) was applied at a 1:500 dilution and HGF antibody (rabbit, clone HPA-044088; Sigma-Aldrich, Burlington, Massachusetts, USA) was applied at a 1:50 dilution for 32 min, followed by the Ventana BenchMark XT automated immunostainer (Ventana, Tucson, AZ, USA) standard Cell Conditioner 1 (CC1, a proprietary buffer) at 95 °C for 8 min (TGF β 1) and Cell Conditioner 2 (CC2, a proprietary buffer) at 91 °C for 8 min (HGF). Finally, the sections were developed in DAB at 37 °C for 8 min, counterstained with Mayer’s hematoxylin, and mounted [14].

Digital images of stained slides were captured using the Aperio ScanScope XT Slide Scanner (Aperio Technologies, Vista, CA, USA) under 20× objective magnification (0.5 µm resolution) for analysis.

### 2.6. Study Outcomes and Statistical Analysis

All participants were grouped into a (1) CRLM or (2) control group based on the type of liver tumor. The main study outcome was regenerated liver volume on POD 30. The impact of TGF β 1 and HGF, together with other clinical variables, on liver regeneration was evaluated. Resected liver volume was defined as the difference between liver volume on baseline and POD 1. Regenerated liver volume was defined as the difference between liver volume on POD 30 and POD 1.

All statistical analyses were conducted using the statistical program SPSS 24.0 (SPSS, Chicago, IL, USA). Continuous variables were expressed as mean and standard deviation (SD). Categorical variables are shown as proportions. An independent-samples *t*-test was used to compare continuous variables. Fisher’s exact test or Pearson’s chi-square test was used to analyze categorical variables, as appropriate. For correlation, Pearsons’s method was used. For multivariable analysis of variables’ impact on liver regeneration, linear regression was performed. A *p*-value < 0.05 was considered statistically significant. GraphPad Prism 8.0 (GraphPad Software Inc., San Diego, CA, USA) was used for graphs.

## 3. Results

### 3.1. Patients

In total, 34 patients (17 in each group) were enrolled in the study. The baseline clinicopathologic characteristics were similar between the control and CRLM groups (Table 1). Indications for liver resection in the control group were echinococcosis (n = 6, 35.3%), cystadenoma (n = 5, 29.4%), follicular nodal hyperplasia (n = 1, 5.8%), hemangioma (n = 2, 11.9%), PEComa (n = 1, 5.8%), and liver cyst (n = 2, 11.9%). In the CRLM group, all patients received preoperative chemotherapy. The exact chemotherapy regimens used for first-line treatment are shown in Table 1. Four (23.5%) patients received second-line chemotherapy with capecitabine (n = 3; 17.6%) or FOLFOX + immunotherapy (n = 1; 5.9%).

### 3.2. Liver Resection

The baseline total liver volume was similar between the control and CRLM groups, 1312 ± 505 mL vs. 1168 ± 321 mL (*p* = 0.327). The most common type of surgery was atypical resection, and the mean resected liver volume was 312 ± 305 mL and 223 ± 307 mL in the control and CRLM groups, respectively (*p* = 0.401) (Table 2). Postoperative complications occurred in three (17.6%) patients in each study group. One (5.9%) patient in each group had postoperative bilioma, and one (5.9%) patient suffered a surgical site infection. Additionally, in the control group, one (5.9%) patient developed a biliary fistula, and one (5.9%) patient in the CRLM group suffered postoperative ascites.

### 3.3. Postoperative Liver Regeneration

The mean liver regeneration volume within 30 days postoperatively was significantly higher in the control group (200 ± 180 mL) compared to the CRLM group (72 ± 154 mL); *p* = 0.03. Liver functions representing liver enzymes (AST, ALT, GGT), albumin, SPA, and HGF levels were similar between the control and CRLM groups before surgery and on POD 1, 3, and 7. Baseline AP and TGF-β1 blood levels were significantly higher in the CRLM group. Liver stiffness was increased in the CRLM group on POD 1 (Figure 1). Immunohistochemical analysis confirmed that a higher proportion of patients in the CRLM group presented high expression of TGF-β1 in liver tissues compared to the control group (*p* = 0.034) (Figure 2).

### 3.4. Biomarkers for Liver Regeneration

Baseline liver stiffness by elastography (R = −0.212; *p* = 0.22), plasma TGF (R = −0.198; *p* = 0.26), AST (R = 0.229; *p* = 0.19), ALT (R = 0.241; *p* = 0.16), GGT (R = 0.033; *p* = 0.85), SPA (R = −0.234; *p* = 0.18), and bilirubin (R = 0.044; *p* = 0.80) levels did not correlate with postoperative liver regeneration. In contrast, resected liver volume (R = 0.568; *p* = 0.01), baseline plasma HGF (R = 0.358; *p* = 0.03), AP (R = −0.361; *p* = 0.03), and albumin (R = 0.339; *p* = 0.05) levels had a significant correlation with liver regeneration. Subsequent multivariable linear regression showed that only resected liver volume (β: 0.31; 95% CI: 0.14–0.47, *p* = 0.01) significantly impacted liver regeneration (Table 3).

## 4. Discussion

This longitudinal observational study showed that CRLM patients with previous chemotherapy treatment have compromised liver regeneration capacity compared to those undergoing liver resections for benign pathology. Individuals with CRLM and previous chemotherapy treatment exhibit lower levels of TGFβ-1 in their blood and reduced TGF-β1 expression in the liver tissue. Thus, it can be assumed that TGF-β1 may be involved in mechanisms that hinder the capacity of liver regeneration following major resection after chemotherapy.

Recent advancements in surgery and chemotherapy have significantly improved the survival rates of patients with metastatic liver cancer, particularly those with CRLM [15]. Surgical resection continues to be the preferred treatment approach for isolated CRLM, demonstrating 5- and 10-year survival rates of approximately 40% and 25%, respectively [16]. While modern surgical techniques and effective intraoperative bleeding control contribute to preventing postoperative liver failure, emerging evidence suggests that the microstructure of liver parenchyma also plays a crucial role in both liver regeneration and the avoidance of postoperative liver failure [6,17,18,19]. As a result, currently, there is a strong recommendation for a meticulous selection process when choosing patients for major liver resection. It is advised to thoroughly assess liver volume and function to minimize the risk of postoperative liver failure, with a special emphasis on maintaining at least 30% of the liver as a future liver remnant, especially for patients after chemotherapy [5,20,21]. However, some conflicting evidence exists regarding chemotherapy’s impact on liver regeneration. A recent meta-analysis suggests that neoadjuvant chemotherapy has no negative impact on post-hepatectomy liver regeneration [9,22]. Such findings contradict our current study, which revealed a significant decrease in regenerated liver volume among patients with CRLM who underwent chemotherapy when compared to controls who underwent surgery for benign pathology. The differences between the previous and current findings may be attributed to the robust methodology of the present study. First, we longitudinally observed the patients and measured liver volume at baseline, on POD 1 and POD 30. Second, we had a control group that consisted of chemotherapy treatment-naïve patients with benign pathology.

The process of the restoration of liver volume involves the replication of different intrahepatic cell types, accompanied by an enlargement of cell size. The timing of hepatocyte replication onset and its peak varies across species [4]. In humans, following partial hepatectomy, intracellular signaling pathways in hepatocytes are rapidly activated within minutes, although the specific mechanisms triggering these pathways remain unclear [7]. The primary mechanism for liver regeneration after both acute and chronic liver injury appears to be the compensatory proliferation of remaining hepatocytes. Moreover, various studies have acknowledged the occurrence of transdifferentiation between hepatocytes and cholangiocytes [23].

The exact pathological mechanisms of how chemotherapy impairs liver regeneration remain unknown, but HGF and TGF-β1 may be involved in the initiation and termination of regeneration [11,24,25]. These factors’ role has been widely investigated in preclinical research, but there is a lack of evidence from clinical studies. It is considered that HGF participates in liver regeneration in the proliferation stage by inducing mitosis [26]. This event happens as a follow-up to increased portal blood flow to remaining segments and might increase liver stiffness [27]. Once an adequate liver volume is achieved, TGF-β1 acts as a brake on the regeneration process by downregulating HGF synthesis [28]. In the present study, we investigated these molecules in a clinical setting. Our experiments showed for the first time that CRLM patients treated with chemotherapy have lower TGF-β1 serum levels and TGF-β1 expression in liver tissue by immunohistochemistry. Further, we showed increased liver stiffness in the CRLM group on POD 1, confirming previous evidence that the start of liver regeneration is not mediated through edema [29].

Biomarkers for postoperative liver regeneration may improve CRLM treatment outcomes because they would allow a personalization of surgical treatment and may avoid omitting the surgery in patients with borderline insufficient remnant liver volume. Liver function representing enzymes has been investigated as a potential biomarker for regeneration, and some combinational scores are available [30,31,32]. The current study investigated a panel of liver enzymes, albumin, SPA, liver stiffness by elastography, and serum HGF and TGF-β1 correlation with liver regeneration and found that resected liver volume, baseline serum HGF, alkaline phosphatase, and albumin levels have a moderate correlation. However, subsequent multivariable analysis showed only resected liver volume as an independent predictor for liver regeneration. Of course, the present study has a relatively small sample size; thus, it may be underpowered for biomarker analysis.

The current study has some limitations. First, this study is not a randomized trial, due to the nature of the selection criteria and specific pathology; multicenter randomized trials might provide more in-depth evidence. Secondly, HGF and TGF-β1 are only a part of a multiprotein family involved in mechanisms in liver regeneration. Thirdly, it needs to be further investigated whether HGF and TGF-β1 tissue expression is associated with induction by chemotherapy or tumor burden.

## 5. Conclusions

This study showed that CRLM patients with previous chemotherapy treatment have compromised liver regeneration capacity compared to patients with benign pathology. Individuals with CRLM and previous chemotherapy treatment exhibit lower serum TGF-β1 levels and reduced TGF-β1 expression in the liver tissue. Thus, it can be assumed that TGF-β1 may be involved in mechanisms that hinder the capacity of liver regeneration following major resection after chemotherapy.

## Figures and Tables

**Figure 1 jpm-14-00144-f001:**
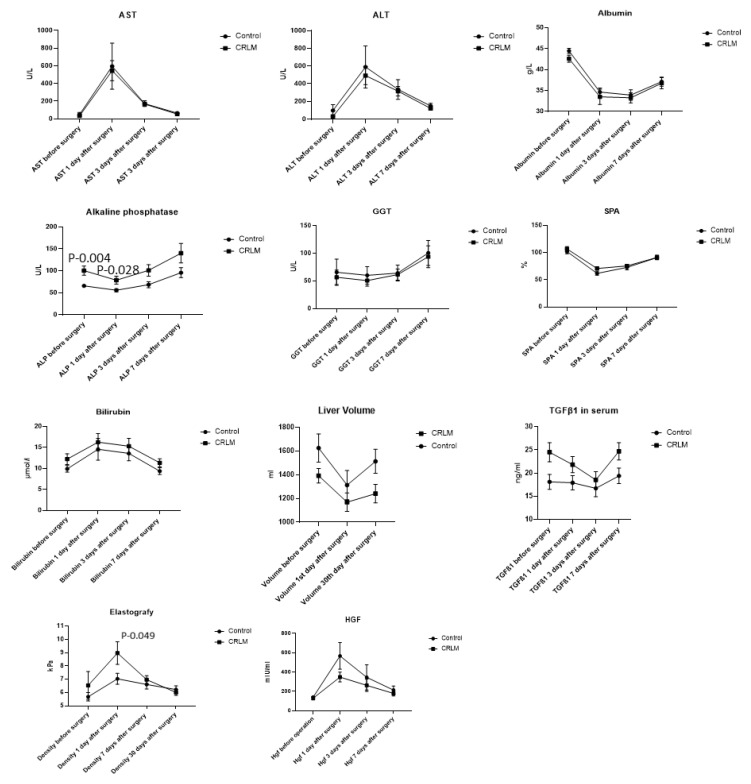
Dynamics of static liver function enzymes, liver stiffness, liver volume, and regeneration marker expression. Data are represented as means with standard deviation. Significance was achieved when *p* < 0.05.

**Figure 2 jpm-14-00144-f002:**
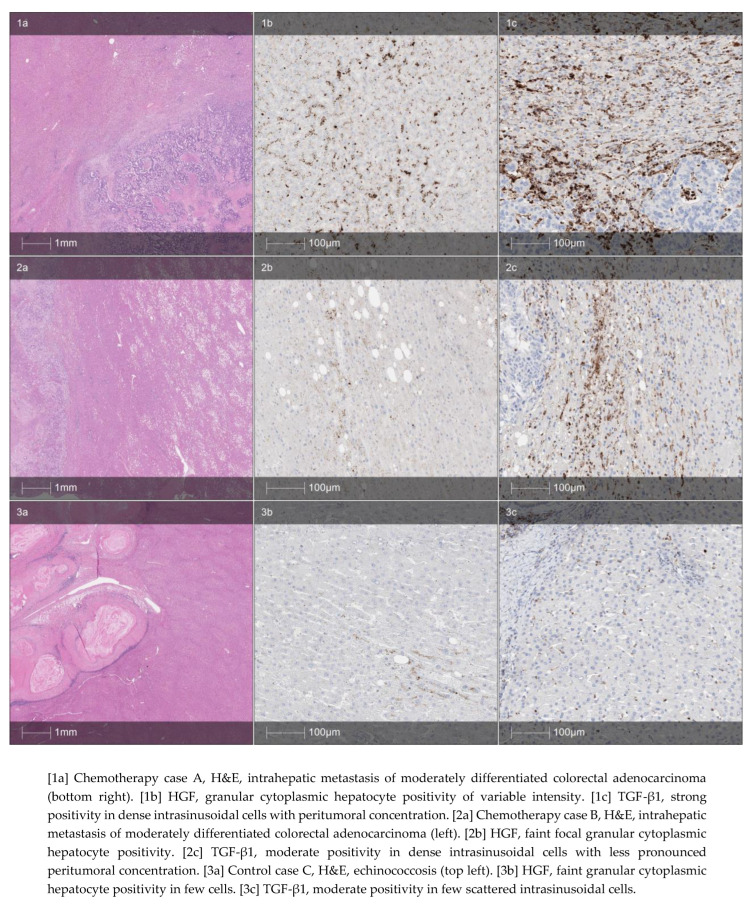
Sample histology (hematoxylin–eosin, column (a)) and immunohistochemistry (HGF, column (b) and TGF, column (c)). 1 and 2—chemotherapy cases; 3—control group case. Graph of HGF and TGF-β1 intensity distribution among groups.

**Table 1 jpm-14-00144-t001:** Baseline clinicopathologic features of individuals who had liver resections in the control and CRLM groups. Data are shown as means with standard errors. Significance was considered when *p* < 0.05.

		Control Group (n = 17)	CRLM Group (n = 17)	*p*-Value
Age (years), mean (SD)		50 ± 14.	59 ± 12	0.07
Gender, M/F		5/12	5/12	0.99
BMI (kg/m^2^), mean (SD)		30.7 ± 7.3	25.6 ± 5.9	0.07
Comorbidities, n (%)	Cardiovascular	2 (11.7%)	9 (52.9%)	0.06
	Diabetes	2 (11.7%)	0 (0%)	0.48
	Other	1 (5.8%)	0 (0%)	0.99
Primary tumor site, n (%)	Cecum	N/A	2 (11.9%)	N/A
	Hepatic flexure		2 (11.9%)	
	Sigmoid colon		7 (41.2%)	
	Rectum		6 (35.3%)	
Number of metastases, n (%)	1	N/A	11 (64.7%)	N/A
	2–5		4 (23.5%)	
	>5		2 (11.8%)	
Type of metastases, n (%)	Synchronous	N/A	5 (29.4%)	N/A
	Metachronous		12 (70.6%)	
Preoperative CEA level (ng/mL), mean (SD)		N/A	44 ± 167	N/A
Preoperative CA 19.9 (U/mL), mean (SD)		N/A	37 ± 78	N/A
Median number of preoperative chemotherapy cycles (Q1–Q3)		N/A	8 (6; 12)	N/A
Type of first-line preoperative chemotherapy, n (%)	FOLFOX	N/A	5 (29.4%)	N/A
	FOLFOX + bevacizumab	N/A	7 (41.2%)	N/A
	XELOX	N/A	1 (5.9%)	N/A
	XELOX + bevacizumab	N/A	2, (11.9%)	
	Capecetabin	N/A	2, (11.9%)	

**Table 2 jpm-14-00144-t002:** Characteristics of metastasis and treatment in the chemotherapy group. Data are shown as means with standard error or percentage.

		Control Group (n = 17)	CRLM Group (n = 17)	*p*-Value
Type of liver resection, n (%)	Atypical	8 (47.1%)	11 (64.7%)	0.21
	Left lobectomy	4 (23.5%)	3 (17.6%)	
	Right lobectomy	1 (5.9%)	0 (0%)	
	Left hepatectomy	1 (5.9%)	1 (5.9%)	
	Right hepatectomy	3 (17.6%)	0 (0%)	
	ALPPS	0	2 (11.9%)	
Length of surgery (minutes), mean (SD)		181 ± 64	251 ± 168	0.12
Blood loss (mL), mean (SE)		520 ± 552	516 ± 648	0.98
Patients requiring postoperative blood transfusions, n (%)		6 (35.3%)	5 (29.4%)	0.58
Postoperative hemotransfusion units, mean (SD)		0.53 ± 1.12	0.76 ± 1.30	0.58

**Table 3 jpm-14-00144-t003:** Multivariable analysis of factors for postoperative liver regeneration.

Variable	β Coefficient (95% Confidence Interval)	*p*-Value
Baseline HGF value	0.52 (−0.27–1.31)	0.189
Baseline albumin value	7.11 (−10.01–23.24)	0.403
Baseline alkaline phosphatase value	−1.403 (−2.95–0.14)	0.074
Resected liver volume	0.31 (0.14–0.47)	0.001

95% CI: 95% confidence interval.

## Data Availability

The datasets used and/or analyzed during the current study are available from the corresponding author upon reasonable request.

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
