# Peer review of "The Impact of Chemotherapy and Transforming Growth Factor-β1 in Liver Regeneration after Hepatectomy among Colorectal Cancer Patients"

_jpm, 2024, doi:10.3390/jpm14020144_

Round 1
Reviewer 1 Report
Comments and Suggestions for Authors
Thanks for inviting me to evaluate the article titled ‘The Impact of Chemotherapy and Transforming Growth Factor-β1 in Liver Regeneration After Hepatectomy among Colorectal Cancer Patients’. In this paper, the authors demonstrated, this study focused on liver regeneration in CRLM patients’ post-chemotherapy. Additionally, they exhibited lower serum TGFβ-1 levels and reduced TGFβ-1 expression in liver tissue, suggesting TGFβ-1 involvement in mechanisms hindering liver regeneration capacity following major resection after chemotherapy. This paper is of novelty. While some parts of the text still need to be revised. Figure 2 is hard to see. Please provide a clear one. So, I recommend reconsidering this paper after minor revision.
Author Response
Dear reviewer, thank you for your response. High-resolution images will be provided in supplementary material.
Reviewer 2 Report
Comments and Suggestions for Authors
Dear authors,
Thanks for your outstanding work
However, there are a few remarks to be noted:
- Minor comments
1- What was that indication of hepatectomy in focal nodular hyperplasia?
N/B: It is "focal" not "follicular" as written by mistake in the manuscript
2- The resolution of the written text in Figure two must be improved
- Major comments:
3 - Please perform deeper statistical analysis if a certain chemotherapy regimen affected liver regeneration more than the other types
4- please mention the used immunotherapy types and perform statistical analysis also to see if a certain type has more affection than the others
Author Response
Dear reviewer, thank you for your time and comments regarding manuscript.
1 - spelling mistake was corrected;
2 - resolution of text and figure was improved;
3 - deeper statistical analysis is not possible due to fairly small amount of cases;
4 - immunotherapy regimens were specified